# ResVG: Enhancing Relation and Semantic Understanding in Multiple Instances for Visual Grounding

## ABSTRACT

Visual grounding aims to localize the object referred to in an image based on a natural language query. Although progress has been made recently, accurately localizing target objects within multiple-instance distractions (multiple objects of the same category as the target) remains a significant challenge. Existing methods demonstrate a significant performance drop when there are multiple distractions in an image, indicating an insufficient understanding of the fine-grained semantics and spatial relationships between objects. In this paper, we propose a novel approach, the **Re**lation and **S**emantic-sensitive **V**isual **G**rounding (ReSVG) model, to address this issue. Firstly, we enhance the model's understanding of fine-grained semantics by injecting semantic prior information derived from text queries into the model. This is achieved by leveraging text-to-image generation models to produce images representing the semantic attributes of target objects described in queries. Secondly, we tackle the lack of training samples with multiple distractions by introducing a relation-sensitive data augmentation method. This method generates additional training data by synthesizing images containing multiple objects of the same category and pseudo queries based on their spatial relationships. The proposed ReSVG model significantly improves the model's ability to comprehend both object semantics and spatial relations, leading to enhanced performance in visual grounding tasks, particularly in scenarios with multiple-instance distractions. We conduct extensive experiments to validate the effectiveness of our methods on five datasets.

## CCS CONCEPTS

• **Computing methodologies → Artificial intelligence**.

## KEYWORDS

Visual grounding, Referring Expressions, Data Augmentation, Stable Diffusion

## 1 INTRODUCTION

Visual grounding[10, 26, 51] aims to localize the object referred to in an image based on the given natural language query. It is a key element in multi-modal reasoning systems, applicable across various tasks such as visual question answering[3] and vision-and-language navigation[2]. Moreover, it serves as a surrogate for assessing machines in open-ended scene recognition and localization. Unlike

**Unpublished working draft. Not for distribution.**

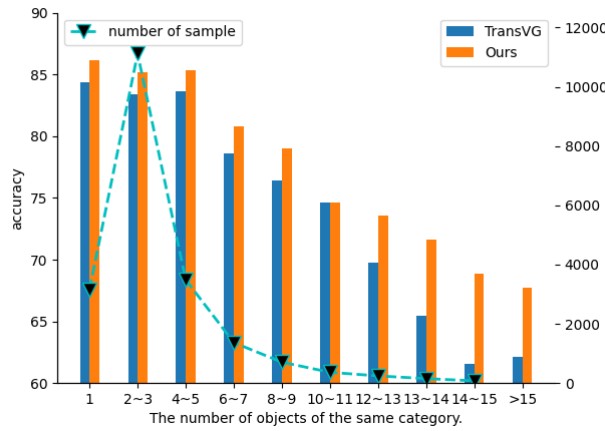

**Figure 1: The performance and number of samples on the RefCOCO dataset when there are different numbers of objects of the same category with the target object in an image.**

conventional object detection methods confined to recognizing pre-defined categories within training data, visual grounding involves pinpointing the referenced object, often specified by one or more pieces of information in the language query. The information contains not only object categories but also appearance attributes, visual relation contexts, etc.

Dealing with multiple instances of distraction, where there are several objects of the same category as the target object within the image, poses a substantial challenge for existing visual grounding models. Accurately locating the target object among multiple instances of the same category requires the model to thoroughly leverage textual information and model distinctive visual features for precise visual grounding. Empirically, as shown in Figure 1, we conducted a statistical analysis of the performance of existing methods when dealing with images containing different numbers of distraction objects (of the same category as the target object) on the RefCOCO dataset. It can be observed that as the number of distraction objects increases, the model's performance exhibits a noticeable decrease. We discuss the plausible reasons as follows.

*Firstly, current models may not have a sufficient understanding of the fine-grained semantics of the target object.* For some samples, despite containing multiple distraction objects of the same category as the target object, their fine-grained appearance attribute semantics may differ (such as color, shape, texture, etc). In such cases, descriptions in queries regarding these fine-grained semantics become crucial (e.g., a person wearing a yellow shirt). What adds to the challenge is that the appearance attribute proving most discriminative for grounding is not solely determined by the fine-grained attribute semantic word itself. Instead, it is also influenced

by the frequency of instances sharing that attribute within the specific given image. However, these fine-grained semantics involve complex combinations of multiple attributes across different categories, posing a challenge for the model to fully comprehend this fine-grained semantic information. *Secondly, in the existing dataset, the distribution of the number of distraction objects exhibits a long-tail distribution, making it difficult for current models to fully understand the spatial relationships between objects.* In Figure 1, we show the distribution for different numbers of distraction objects on the RefCOCO dataset, revealing that a large number of samples only contain less than 3 objects of the same category as the target object. Although for such samples, queries may also involve spatial relationships between objects, the model may only need to locate the target object based on the semantic description of the target object without necessarily understanding the spatial relationships between them. As the number of distraction objects increases, the model needs to understand their spatial relationships more, but these samples are relatively rare, making it difficult for the model to learn.

To tackle the above problems, we propose the **Re**lation and **S**emantic sensitive **V**isual **G**rounding (ReSVG) model to enhance the model's understanding of both object spatial relations and semantics. *Firstly, for the insufficient understanding of fine-grained semantics, we propose to inject the semantic prior information of the target object according to the query for the model.* Specifically, inspired by the powerful generalization ability of existing text-to-image generation models, we propose to use queries as text prompts to generate images of target objects through off-the-shelf text-to-image models, serving as prior information. Typically, the generated image centers on a zoomed-in, object-centric view, providing a clear visual perspective of the intricate details corresponding to the appearance semantics mentioned in the query (such as color, shape, texture, etc), thus we use the generated images as prior information of the target objects to help the model better understand the semantics of the target object. Injecting visual semantic priors can better assist the model in focusing on the correct parts of the input image compared to using textual queries only, as the homogeneous gap within the visual domain is much smaller than the heterogeneous gap between vision and text. *Secondly, to address the problem of insufficient understanding of spatial relationships between objects due to a lack of training samples with multiple distractions, we propose a relation-sensitive data augmentation method to generate more relation-sensitive multiple-distraction data.* Specifically, we use class names with quantity words as text prompts and use text-to-image models to generate images containing multiple objects of the same category. Then we use an object detector to detect the boxes of objects in the images and generate pseudo queries based on the spatial relationships of these boxes. For these generated images and pseudo queries, the model can only accurately locate the target objects if it understands the spatial relationships between them. We train the model with the generated data together with the original dataset to enhance the model's understanding of spatial relationships.

Our contributions are summarized as follows: (1) To help the model understand the fine-grained semantics of the target object, we propose the semantic prior injection which injects the semantic prior information of the target object using text-to-image models

for the visual grounding. (2) To enhance the model's understanding of spatial relationships, we propose a relation-sensitive data augmentation method to generate more relation-sensitive multiple-instances data. The above design significantly improves the performance when facing multiple instances of distraction. (3) We conduct extensive experiments to validate the effectiveness of our methods on the RefCOCO [51], RefCOCO+ [51], RefCOCOg [26], ReferItGame [20] and Fliker30K Entities[29] datasets.

## 2 RELATED WORK

### 2.1 Visual Grounding

Existing methods generally extend the object detection framework to address the visual grounding task by incorporating a visual-linguistic fusion module. The early methods can be categorized into two-stage and one-stage methods. Two-stage methods [16, 17, 23, 42, 43, 47, 50, 53, 57] leverage the off-the-shelf detectors to generate a set of proposals from the image in the first stage, and then match them with the language expression to select the top-ranked proposal. One-stage approaches[8, 18, 22, 48, 49] fuse the visual features and the language-attended feature maps, and output the boxes directly. Recently, transformer-based methods[10, 25, 30, 36, 38, 41, 46, 56] achieve remarkable results on visual grounding. They take the visual and linguistic feature tokens as inputs, feed them into a set of transformer encoder layers to perform cross-modal fusion, and predict the target region directly. Considering their performance advantages, we apply our approach to the Transformer-based method, TransVG [10].

However, existing methods often demonstrate a significant performance drop when there are multiple distractions in an image, indicating an insufficient understanding of the fine-grained semantics and spatial relationships between objects. In this paper, to tackle these aforementioned limitations, we first propose generating high-quality images containing multiple objects to construct pseudo pairs that describe the spatial relationship between objects, thus enriching the model's comprehension of spatial relationships. Additionally, we also propose to generate visual features highly relevant to the textual queries as prior information to help the model better understand the semantics of target objects.

### 2.2 Data Augmentation

In many research fields, collecting and calibrating a large amount of data requires high costs or is even infeasible. The performance of most deep learning methods is strongly dependent on a large amount of data samples with rich feature diversity. A model trained on small-size datasets usually has limited performance. Currently, data augmentation methods have attracted increasing attention. Traditional data augmentation methods[32, 37, 39, 55] include image warping, deformation, random cropping, random flip blurring, image sharpening and blurring, changing color spaces, a weighted sum of two images, and the inverse transformation after adding noise. Recently, generative models [13, 21, 34] showed astonishing results for synthesizing images. Numerous approaches have been published that adapt generative models to synthesizing images for data augmentation. In the field of image classification[5, 9, 27], only a text prompt or category label is needed to generate a large number of images for model training. In some fine-grained tasks such as

image segmentation[1, 14, 40, 44, 45] and object detection[7, 54], existing methods often exploits cross-modal attention maps between image features and conditioning text embedding to obtain pseudo labels for supervision.

To the best of our knowledge, we are the first to utilize the generative model to enhance spatial relationship understanding and semantic sensitive capabilities of the grounding model. Our approach differs from existing works in two key aspects: 1) We leverage the generative model to generate images containing multi-instances, thereby enhancing the comprehensive understanding of spatial relationships.2) We also use input queries as text prompts to generate object-centric images for guiding the model decoding, thereby enhancing the model's semantic-sensitive capabilities (e.g., appearance attributes).

## 3 METHOD

**Task formulation.** Given an image $I$ and a text query $Q$, the visual grounding task requires the model to output the bounding box $b = (x, y, w, h)$ of the target object referred to in the image $I$ based on the query $Q$.

In this section, we first revisit our baseline TransVG[10] in Section 3.1. Then, we give an overview of our proposed model in Section 3.2. Finally, we give the details of our relation-sensitive data augmentation in Section 3.3 and our semantic-sensitive visual grounding in Section 3.4.

### 3.1 TransVG Revisited

TransVG[10] serves as our baseline model, consisting of four main modules as shown in Figure 2: a text encoder, a visual encoder, a transformer decoder, and a prediction head.

**Text encoder:** TransVG takes the query $Q$ as input and uses the pre-trained BERT[11] as the text encoder to extract textual embeddings of each token $F_l = [p_l^1, p_l^2, ..., p_l^{N_l}] \in \mathbb{R}^{N_l \times D}$, where $N_l$ is the maximum length of the text query and $D$ is the feature dimension.

**Visual encoder:** TransVG uses the ResNet[15] as the backbone to generate 2D visual feature map $z \in \mathcal{R}^{D \times H \times W}$ from the input image $I$, where $H$ and $W$ are the width and height of the feature map respectively. Then, TransVG further flattens $z$ and uses a Transformer encoder to extract visual semantic information $F_v = [p_v^1, p_v^2, ..., p_v^{HW}] \in \mathbb{R}^{HW \times D}$.

**Transformer decoder:** TransVG uses a transformer decoder to fuse the linguistic and visual features and decode the target object. Given the visual and textual embeddings $F_v, F_l$ outputted from the visual and text encoders, they are concatenated and padded with a learnable target query $p_r$. Then, the transformer decoder performs self-attention between the concatenated tokens to decode the target object feature $\hat{p}_r$:

$$\hat{p}_r, \hat{F}_l, \hat{F}_v = \mathbf{D}([p_r; F_l; F_v]), \tag{1}$$

where $\mathbf{D}(\cdot)$ is the transformer decoder and $[;]$ represents the feature concatenation.

**Prediction Head:** Finally, the prediction head takes the target object feature $\hat{p}_r$ as input and uses a multilayer perceptron (MLP) to predict the box coordinates $b$ of the target object. The final loss

function is:

$$\mathcal{L} = \mathcal{L}_{smooth-l1}(b, \hat{b}) + \mathcal{L}_{giou}(b, \hat{b}), \tag{2}$$

where $\hat{b}$ is the ground-truth box, $b$ is the predicted box, $\mathcal{L}_{smooth-l1}$ is the Smooth L1 loss, and $\mathcal{L}_{giou}$ is the GIoU loss [33].

### 3.2 Overview

Our approach, as illustrated in Figure 2, mainly consists of two parts: Relation-Sensitive Data Augmentation and Semantic-Sensitive Visual Grounding. Firstly, to address the problem of insufficient understanding of spatial relationships between objects due to a lack of training samples with multiple distractions, we propose a relation-sensitive data augmentation method to generate more relation-sensitive multiple-distraction data. Specifically, we utilize class names with quantity words as text prompts and employ text-to-image models to generate images containing multiple objects of the same category. Subsequently, an object detector is used to detect the boxes of objects in these images and pseudo queries are generated based on the spatial relationships of these boxes. For these generated images and pseudo queries, the model can accurately locate the target objects only if it comprehends the spatial relationships between them. We train the model with the generated data alongside the original dataset to enhance the model's understanding of spatial relationships. Secondly, to facilitate the model in understanding the fine-grained semantic information of the target objects, we inject semantic priors into the baseline grounding model TransVG. Specifically, we propose to utilize a text-to-image model with queries as text prompts to generate images of the target objects, incorporating fine-grained semantic information such as color and shape. We encode the generated prior images using an image encoder and add them with the learnable token in the baseline, thus injecting the semantic priors of the target objects into the baseline.

### 3.3 Relation-Sensitive Data Augmentation

To enable existing methods to fully understand descriptions of object relationships in queries, we propose Relation-Sensitive Data Augmentation, synthesizing data containing multiple distractions that requires understanding descriptions of relationships between objects in queries to accurately locate them for model training.

Specifically, inspired by the outstanding performance of the text-to-image models (such as Stable Diffusion[34]), we propose utilizing the powerful image generation capability of the Stable Diffusion model to generate high-quality images with multiple objects with the same categories. As shown in Figure 2, for each object category, we randomly generate several quantity words and use each quantity word followed by the category name as the text prompt to generate images using stable diffusion[34]. For example, as shown in Figure 2, using 'three orange' as the text prompt can generate images containing three instances of oranges. After obtaining high-quality images with multiple instances of the same class objects, we detect the bounding boxes of each object in the images using an off-the-shelf object detector and utilize the method in CPL [24] to generate some pseudo-queries for each object for training. Specifically, we predefine a set of spatial relationships (e.g. top, bottom, front, behind, left top, leftmost, second right, etc.), and then determine the spatial relation of each object by comparing the center

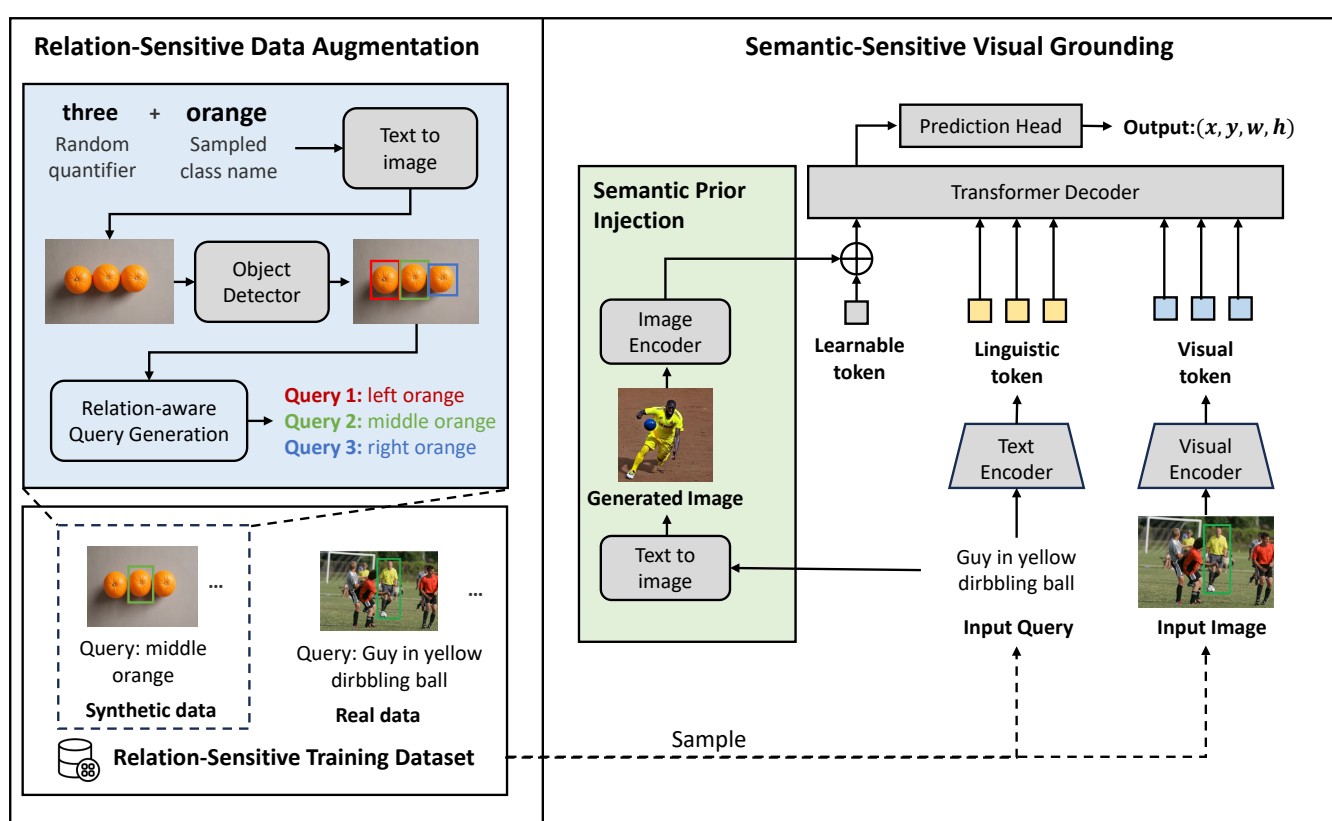

Figure 2: Our method comprises two key components: Relation-Sensitive Data Augmentation and Semantic-Sensitive Visual Grounding. Firstly, we augment training data with multiple instances of the same category, emphasizing spatial relationships through generated images and pseudo queries. Secondly, we inject fine-grained semantic information into the grounding model to enhance understanding of object semantics.

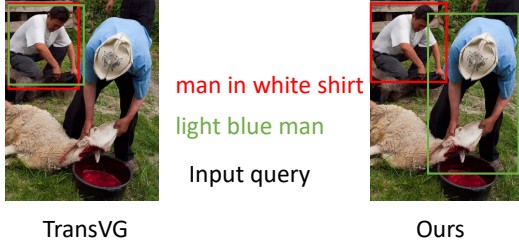

Figure 3: Comparasion with the baseline TransVG [10]. Our method can better distinguish target objects of the same category but with different fine-grained attribute semantics.

coordinates and area of each object box output by the detector. Finally, we obtain the pseudo queries based on the template '{Rela} {Noun}', such as 'middle orange'.

In the generated images, there are multiple instances of the same category. To accurately locate one of them, the model needs to fully understand the spatial relationships described in the pseudo-queries. Therefore, we blend these synthetic data with real data to create a relationship-sensitive training dataset, enabling the model to learn the spatial relationships between objects thoroughly.

## 3.4 Semantic-Sensitive Visual Grounding

For some samples, despite containing multiple distraction objects of the same category, their fine-grained appearance attribute semantics may differ (such as color, shape, texture, etc.), and descriptions in queries regarding these fine-grained semantics become crucial. In the baseline, a single global shared learnable token is used to decode features of the target objects for all queries. However, different queries describe different target objects, making it difficult to learn the prior information of various target objects with different attribute semantics using the token shared by all queries. As shown in Figure 3, the baseline model fails to distinguish between the two objects that have the same category but different fine-grained attributes. Therefore, we propose using the Stable Diffusion model to obtain additional semantic prior information corresponding to the query and combine the semantic prior with the learnable token in the baseline. This makes the learnable token semantic-aware and can guide the model to focus on region features that are visually more similar to the prior, thus achieving more accurate localization as shown in Figure 3.

| Method | Backbone | RefCOCO | | | RefCOCO+ | | | RefCOCOg | | |
|--------|----------|-----|-------|-------|-----|-------|-------|-------|-------|--------|
| | | val | testA | testB | val | testA | testB | val-g | val-u | test-u |
| *Two-stage:* | | | | | | | | | | |
| CMN | VGG16 | - | 71.03 | 65.77 | - | 54.32 | 47.76 | 57.47 | - | - |
| VC | VGG16 | - | 73.33 | 67.44 | - | 58.40 | 53.18 | 62.30 | - | - |
| ParalAttn | VGG16 | - | 75.31 | 65.52 | - | 61.34 | 50.86 | 58.03 | - | - |
| MAttNet | ResNet-101 | 76.65 | 81.14 | 69.99 | 65.33 | 71.62 | 56.02 | - | 66.58 | 67.27 |
| LGRANs[43] | VGG16 | - | 76.60 | 66.40 | - | 64.00 | 53.40 | 61.78 | - | - |
| DGA[47] | VGG16 | - | 78.42 | 65.53 | - | 69.07 | 51.99 | - | - | 63.28 |
| RvG-Tree[16] | ResNet-101 | 75.06 | 78.61 | 69.85 | 63.51 | 67.45 | 56.66 | - | 66.95 | 66.51 |
| NMTree[23] | ResNet-101 | 76.41 | 81.21 | 70.09 | 66.46 | 72.02 | 57.52 | 64.62 | 65.87 | 66.44 |
| Ref-NMS[6] | ResNet-101 | 80.70 | 84.00 | 76.04 | 68.25 | 73.68 | 59.42 | - | 70.55 | 70.62 |
| *One-stage:* | | | | | | | | | | |
| SSG[8] | DarkNet-53 | - | 76.51 | 67.50 | - | 62.14 | 49.27 | 47.47 | 58.80 | - |
| FAOA[49] | DarkNet-53 | 72.54 | 74.35 | 68.50 | 56.81 | 60.23 | 49.60 | 56.12 | 61.33 | 60.36 |
| RCCF[22] | DLA-34 | - | 81.06 | 71.85 | - | 70.35 | 56.32 | - | - | 65.73 |
| ReSC-Large[48] | DarkNet-53 | 77.63 | 80.45 | 72.30 | 63.59 | 68.36 | 56.81 | 63.12 | 67.30 | 67.20 |
| LBYL-Net[18] | DarkNet-53 | 79.67 | 82.91 | 74.15 | 68.64 | 73.38 | 59.49 | 62.70 | - | - |
| *Trans-based:* | | | | | | | | | | |
| TransVG[10] | ResNet-50 | 80.32 | 82.67 | 78.12 | 63.50 | 68.15 | 55.63 | 66.56 | 67.66 | 67.44 |
| **TransVG+ours** | ResNet-50 | **82.60** | **85.36** | **78.97** | **66.13** | **70.95** | **62.06** | **67.10** | **68.39** | **69.77** |
| VLTVG [46] | ResNet-50 | 84.53 | 87.69 | 79.22 | 73.60 | 78.37 | 64.53 | 72.53 | 74.90 | 73.88 |
| **VLVTG+ours** | ResNet-50 | **85.51** | **88.76** | **79.93** | **73.95** | **79.53** | **64.88** | **73.13** | **75.77** | **74.53** |

**Table 1: Comparisons with state-of-the-art methods on RefCOCO [51], RefCOCO+ [51], RefCOCOg [26] in terms of top-1 accuracy(%).**

| Method | Backbone | ReferIt | Flickr30K |
|--------|----------|---------|-----------|
| *Two-stage:* | | | |
| CMN[17] | VGG16 | 28.33 | - |
| VC[53] | VGG16 | 31.13 | - |
| MAttNet[57] | ResNet-101 | 29.04 | - |
| SimilarNet[42] | ResNet-101 | 34.54 | 60.89 |
| CITE[28] | ResNet-101 | 35.07 | 61.33 |
| DDPN[52] | ResNet-101 | 63.00 | 73.30 |
| *One-stage:* | | | |
| SSG[8] | DarkNet-53 | 54.24 | - |
| ZSGNet[35] | ResNet-50 | 58.63 | 63.39 |
| FAOA[49] | DarkNet-53 | 60.67 | 68.71 |
| RCCF[22] | DLA-34 | 63.79 | - |
| ReSC-Large[48] | DarkNet-53 | 64.60 | 69.28 |
| LBYL-Net[18] | DarkNet-53 | 67.47 | - |
| *Trans-based:* | | | |
| TransVG[10] | ResNet-50 | 69.76 | 78.47 |
| **TransVG+ours** | **ResNet-50** | **71.01** | **79.02** |
| VLTVG[46] | ResNet-50 | 71.60 | 79.18 |
| **VLVTG+ours** | ResNet-50 | **72.35** | **79.52** |

**Table 2: Comparison with state-of-the-art methods on Refer-ItGame and Flickr30K Entities datasets in terms of top-1 accuracy (%).**

Specifically, to obtain semantic prior for textual queries, we feed input textual queries as prompts into the Stable Diffusion model to generate high-quality images that conform to the input textual queries:

$$\hat{I} = \mathbf{SD}(Q) \tag{3}$$

where $Q$ is the input query, $\hat{I}$ is the generated prior image, and $\mathbf{SD}(\cdot)$ is the Stable Diffusion model. Thanks to the powerful generalization ability of the Stable Diffusion model, the generated image $\hat{I}$ effectively exhibits fine-grained semantic information about the target objects (such as color, shape, texture, etc.) as shown in Figure 2.

Then, to inject the semantic prior in $\hat{I}$ into the model, we encode $\hat{I}$ as a feature using a pre-trained image encoder[31] and add it with the learnable token $p_r$ in baseline to obtain the semantic-aware token $p_f$:

$$p_f = \mathbf{E}(\hat{I}) + p_r \tag{4}$$

Finally, we follow our baseline to use a transformer decoder to decode the feature of the target object feature $\hat{p_f}$ similar to Equation (1). The only difference is we replace the global shared learnable token $p_r$ in Equation (1) with our semantic-aware token $p_f$:

$$\hat{p_f}, \hat{F_l}, \hat{F_v} = \mathbf{D}([p_f; F_l; F_v]) \tag{5}$$

Then, we use a prediction head to predict the target box $b$ the same loss function as the baseline shown in Equation (2) to train the model.

Compared to the baseline that decodes object features for all textual queries using only a shared learnable token, our approach injects the semantic prior of the target object to the shared token, and the obtained semantic-aware token is independent for each sample, allowing better learning of the fine-grained semantics of different target objects. In addition, compared to the baseline that randomly initializes the learnable token for decoding object features,

| Method | val | testA | testB |
|---|---|---|---|
| Baseline(TransVG) | 63.50 | 68.15 | 55.63 |
| +Data Augmentation | 64.77 | 68.56 | 57.54 |
| +semantic-sensitive | 65.75 | 70.07 | 58.58 |
| +Both | **66.13** | **70.95** | **62.06** |

Table 3: Ablations of each component on RefCOCO+ dataset.

| Design | val | testA | testB |
|---|---|---|---|
| Learnable query | 63.50 | 68.15 | 55.63 |
| Image feature | 64.05 | 66.13 | 55.60 |
| Ours | **66.13** | **70.95** | **62.06** |

Table 4: Ablative experiments to study the final target token design in our framework on RefCOCO+ dataset.

our method, by providing visual priors highly aligned with textual queries, can guide the model to focus on region features with high visual similarity, thus achieving more accurate localization.

### 3.5 Discussion

For the relation-sensitive data augmentation, we find that although the stable diffusion model can generate high-quality images containing multiple objects of the same class, it does not always adhere to the quantifier words in our prompt. However, our pseudo-query generation does not rely on the accuracy of the number of objects in the image. Even in such cases, our method can still generate correct pseudo-queries. Some visualizations are provided in Figure 5.

For enhancing the model's understanding of fine-grained semantics, we do not opt for data augmentation methods similar to spatial relations. This is because we find that the multiple instances generated by stable diffusion have similar fine-grained attributes as shown in Figure 6. Therefore, it is difficult to construct pseudo queries that can distinguish them solely based on fine-grained semantics.

For the semantic prior injection, the generated image centers on a zoomed-in, object-centric view, providing a clear visual perspective of the intricate details corresponding to the appearance semantics mentioned in the query. When the query describes distinctive attributes of the target object, it can provide useful prior information to the model. However, we also found that when the target object lacks distinctive attributes (e.g., the three oranges in Figure 2), the prior information may not be helpful, which is a limitation of our method.

## 4 EXPERIMENT

### 4.1 Datasets

We evaluate our methods on five challenging visual grounding benchmarks as follow:

**RefCOCO/RefCOCO+/RefCOCOg:** 1) RefCOCO [51] contains 19,994 images with 50,000 referred objects. Each object has more than one representation, and the entire dataset has a total of 142,210 referring expression divided into training set, validation set, testA set and testB set. 2) RefCOCO+ [51] contains 19,992 images with 49,856 referred objects and 141,564 referring expressions. It is also split into training set, validation set , testA set and testB. 3) Ref-COCOg [26] has 25,799 images with 49,856 referred objects and expressions.There are two split protocols (umd and google) for this data, and we evaluate on both umd protocols (divided into training set, validation set and test set) and google protocols (divided into traing set and validation set) for a comprehensive comparison of our approach.

**ReferItGame:** ReferItGame contains 20,000 images collected from the SAIAPR-12 dataset [12]. We follow the previous works [10, 46] to split the dataset into three subsets, including a train set (54,127 referring expressions), a validation set (5,842 referring expressions), and a test set (60,103 referring expressions).

**Flickr30K Entities:** Flickr30k Entities contains 31,783 images with 427k referred expressions. We follow the same split as in works [10, 46] for train, validation and test subset.

### 4.2 Implementation Details

In the relation-sensitive data augmentation module, we employ the Stable Diffusion model[34] to generate 50 images for each category for pseudo-query generation, with the quantifier words in the prompt randomly selected from 3 to 10. The final number of generated training samples is approximately one-third of the original dataset. For a fair comparison, we follow TransVG[10] to initialize the visual encoder with the backbone and encoder of the DETR[4] model and initialize our text encoder with the BERT[11] model. During training, input images are set to 640×640, and the maximum input length for the query is 20. Image augmentation techniques such as random cropping, flipping, etc., are applied to enhance the model's robustness following TransVG. Our whole model is optimized with the Adamw optimizer in an end-to-end manner. The initial learning rate is set to $1 \times 10^{-4}$ except for the text encoder and visual encoder which have an initial learning rate of $1 \times 10^{-5}$. We train our model for 90 epochs with a learning rate dropped by a factor of 10 after 60 epochs.

### 4.3 Comparisons with State-of-the-art Methods

It should be emphasized that the relation-sensitive dataset augmentation and the semantic prior injection proposed in this paper can be applied to different baselines to improve performance. Therefore, to demonstrate the effectiveness and robustness of our proposed method, experiments are conducted on two Transformer-based visual grounding methods (TransVG and VLTVG) using the Relation-sensitive Data Augmentation and Semantic-Sensitive Visual Grounding proposed in this paper, and performance is reported in all five datasets.

Specifically, we report the top-1 accuracy (%) results following previous works [10, 19]. A prediction is considered correct once the Jaccard overlap between the predicted region and the ground-truth box exceeds 0.5.

**RefCOCO/RefCOCO+/RefCOCOg:** As shown in Table 1, we report top-1 accuracy(%) of our method together with other existing one-stage, two-stage, and Transformer-based methods on RefCOCO, RefCOCO+, and RefCOCOg datasets. Firstly, it can be observed that the method proposed in this paper, whether applied

| Number of image per category | val | testA | testB |
|---|---|---|---|
| 0 | 63.50 | 68.15 | 55.63 |
| 30 | 65.97 | 70.36 | 58.28 |
| 50 | **66.13** | **70.95** | **58.58** |
| 100 | 59.28 | 66.38 | 51.37 |

**Table 5: Ablation of the number of generated images per category on RefCOCO+ dataset.**

based on TransVG or VLTVG baseline, can improve overall performance in all partitions of the three datasets. Such experimental results demonstrate the portability and effectiveness of the two modules proposed in this paper. A notable observation is that utilizing the modules proposed in this paper with TransVG as the base framework can lead to performance improvements of up to 2.69%, 4.53%, and 2.33% on the RefCOO, RefCOCO+, and RefCOCOg datasets, respectively. It can be observed that Transformer-based methods significantly outperform all one-stage and two-stage methods in terms of performance in all partitions of the three datasets. We argue that this is because one-stage and two-stage methods both require visual localization tasks based on extracted object candidates or anchors. And this is also due to the powerful cross-modal understanding and fusion capability of Transformers. Remarkably, using VLVTG as the base framework achieves a performance of 88.76% on the testA subset of RefCOCO, surpassing the performance of the original method by 1.07%

**ReferItGame/Flickr30K Entities:** To further validate the effectiveness of our proposed method, we also report experimental performance under two base frameworks(TransVG and VLTVG) and show the comparisons with other existing visual grounding methods on ReferItGame and Flickr30K Entities dataset in Table 2. It is worth noting that our method achieves top-1 accuracy of 72.35% and 79.52% on ReferItGame and Flickr30K Entities, respectively, when using VLVTG as the base framework, surpassing all other visual localization methods and setting new records. Similarly, it can be observed that whether using TransVG or VLTVG as the base framework, our method improves performance, showcasing the portability and effectiveness of our approach. Moreover, Our method applied in the TransVG framework can respectively improve performance by 1.25% and 0.55% on the ReferItGame and Flickr30K Entities datasets. Finally, Transformer-based methods also outperform all one-stage and two-stage methods in terms of performance.

## 4.4 Ablation Study

In this section, we conduct ablation studies of our method applied to the TransVG framework on the RefCOCO+ dataset, examining the effectiveness of each component as well as different designs within each component.

We take the method TransVG [19] as our baseline model in all the following tables unless otherwise specified.

**Effectiveness of each component:** In Table 3, we investigated the effectiveness of two proposed improvement modules: Relation-Sensitive Data Augmentation and Semantic-Sensitive Visual Grounding. According to the experimental results, it is evident

| Method | val | testA | testB |
|---|---|---|---|
| Baseline(TransVG) | 59.37 | 65.65 | 50.14 |
| Ours | **65.96** | **70.48** | **54.71** |

**Table 6: Performance of cross-dataset where models are trained on RefCOCO dataset and tested on RefCOCO+ dataset.**

that both Relation-Sensitive Data Augmentation and Semantic-Sensitive Visual Grounding lead to performance improvements over the baseline method. Additionally, it is noticeable that the strategy of Relation-Sensitive Data Augmentation yields a relatively modest performance improvement compared to the enhancement achieved by Semantic-Sensitive Visual Grounding over the baseline method. We speculate that there might be two reasons: 1) The Relation-Sensitive Data Augmentation strategy is primarily aimed at increasing the number of samples for sparse object categories. However, text queries regarding sparse object categories are already scarce in the test set, resulting in only marginal improvements in performance; 2) The generated images and pseudo-samples exhibit distribution drift from the original dataset, leading to performance bottlenecks. In contrast, the Semantic-Sensitive Visual Grounding module can uniformly enhance the denoising process of the model for all samples, thus leading to a more significant improvement in performance across the entire dataset.

**Design of the final target token:** The design of the target token $p_f$ in Equation (4) used in the transformer decoder is crucial for the quality of the decoded object features. Therefore, we investigated the differences in design methods for the target query in Table 4. It can be observed that both randomly initialized learnable target queries and designs using only generated images as target queries are inferior to the target query design proposed in this paper, which integrates a learnable target query with image features. This is because using only visual features as the target query for denoising inevitably leads to biases in the specific prior context of the visual modality. In contrast, the method proposed in this paper can provide visual prior information to guide the model in accurately locating objects in the visual space to some extent while also mitigating modality biases.

**Number of the generated images per category:** We investigate the impact of generating different numbers of images per category in Table 5. Increasing the number of generated images can produce more pseudo pairs for data augmentation, which boosts the performance of our model, as shown in Table 5. If the number of generated images is too large, it will lead to a decrease in performance. This is because there is a distribution shift between the generated pseudo-queries and real queries, causing excessive involvement of pseudo-queries in training to interfere with the model's learning of the original data distribution. Thus, we utilize the Stable Diffusion model to generate 50 images for each category in the experiment.

**The the performance of queries with and without spatial relationship description** To further validate the effectiveness of the proposed module, we conducted statistical analysis on queries containing spatial relationship descriptions and queries without spatial relationship descriptions on the RefCOCO dataset as shown

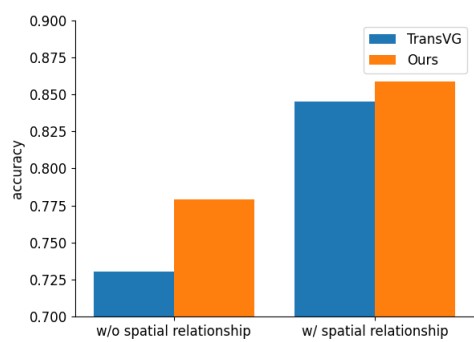

Figure 4: The performance of different queries without and with spatial relationship description on RefCOCO dataset.

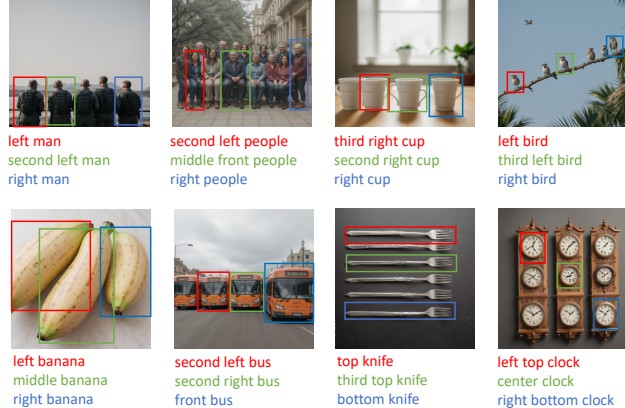

left man
second left man
right man

second left people
middle front people
right people

third right cup
second right cup
right cup

left bird
third left bird
right bird

left banana
middle banana
right banana

second left bus
second right bus
front bus

top knife
third top knife
bottom knife

left top clock
center clock
right bottom clock

Figure 5: Eight visualization examples are generated by the Relation-Sensitive Data Augmentation module.

in Figure 4. It can be observed: 1) for these queries containing spatial relationship descriptions, the proposed data augmentation module can further enhance the model's comprehensive understanding of spatial relationships, 2) for these queries without spatial relationships, the proposed semantic-sensitive visual grounding helps guide the model to achieve more accurate grounding based on the semantic descriptions in the query. Additionally, as shown in Figure 1, our method can effectively enhance the model's understanding and grounding of images containing multiple instances.

**Performance of cross dataset:** We report the performance of our method and TransVG trained on the RefCOCO dataset and tested on the RefCOCO+ dataset in Table 6. It can be observed that the performance of our method is significantly higher than that of TransVG. This experimental result demonstrates that our method can effectively improve the generalization of the model and enhance the model's understanding of fine-grained semantic information. Thanks to the strong generalization capability of our semantic prior injection, even in across dataset scenarios, our method can

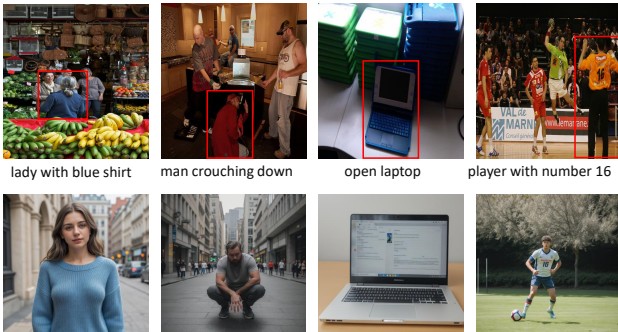

lady with blue shirt    man crouching down    open laptop    player with number 16

Figure 6: Visualizations from the Semantic-Sensitive Visual Grounding. Top: the inputs and predicted bounding boxes. Bottom: the semantic prior image generated from the query.

comprehend fine-grained semantic descriptions in queries to generate the semantic prior and align it with the correct region in the input image. Furthermore, compared to the results on RefCOCO+ in Table 1, both the baseline and our method experienced a decrease in performance, indicating the presence of domain shift between RefCOCO and RefCOCO+. However, the decrease in performance of our model is relatively small (for example, only a 0.47% decrease on the testA split), demonstrating the robustness of our approach.

## 4.5 Qualitative Analysis

To further figure out the importance of spatial relationships, in figure 5, we show eight images generated by the Stable Diffusion model, along with examples generated by the pseudo-query generation module. From the examples, it can be observed that our method can effectively generate high-quality images and pseudo-queries containing spatial relationships for model training, thereby enhancing the model's ability to understand spatial relationships comprehensively. Besides, we also provide examples of images generated by the semantic-sensitive visual grounding module in Figure 6. It can be observed that the generated images are highly semantically correlated with the input text queries and bear a high visual similarity to the ground truth, demonstrating that the proposed method effectively guides the model toward accurate grounding.

## 5 CONCLUSION

In this paper, we propose the Relation and Semantic-sensitive Visual Grounding model to tackle the multiple-instance distractions (multiple objects of the same category as the target) in visual grounding tasks. Existing methods demonstrate a significant performance drop when there are multiple distractions in an image, indicating an insufficient understanding of the fine-grained semantics and spatial relationships between objects. We propose to enhance the model's understanding of fine-grained semantics by injecting semantic prior information derived from text queries into the model and introducing a relation-sensitive data augmentation to address the problem of insufficient understanding of spatial relationships between objects. Experiments on the RefCOCO [51], RefCOCO+ [51], RefCOCOg [26], ReferItGame [20] and Fliker30K Entities[29] datasets demonstrate the effectiveness of our method.

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
