# OpenReview forum: "ResVG: Enhancing Relation and Semantic Understanding in Multiple Instances for Visual Grounding"
_acmmm.org/ACMMM/2024/Conference — MM2024 Poster_

### Official Review · Reviewer_ozsw · 2024-05-23

**Rating:** 5
**Confidence:** 3

**Summary:**

This manuscript introduces the Relation and Semantic-sensitive Visual Grounding (ReSVG) model to address challenges in visual grounding when multiple instances of the same category are present. By injecting semantic prior information and employing relation-sensitive data augmentation, the model aims to improve the comprehension of both object semantics and spatial relationships, leading to enhanced performance in visual grounding tasks.

**Strengths:**

1. This manuscript proposes a novel approach to address the challenge of accurately localizing target objects within multiple-instance distractions by enhancing the model's understanding of fine-grained semantics and spatial relationships.
2. This manuscript introduces two key components: 1) Relation-Sensitive Data Augmentation to generate more relation-sensitive multiple-distraction data, and 2) Semantic-Sensitive Visual Grounding to inject semantic prior information into the model.
3. This manuscript demonstrates strong performance improvements over state-of-the-art methods on five visual grounding datasets.
4. This manuscript provides extensive experiments, ablation studies, and qualitative analysis to validate the effectiveness of the proposed techniques.
5. The proposed model integrates well with existing transformer-based visual grounding models.

**Limitations:**

1. This manuscript could provide more details on the specific architecture and implementation of the ReSVG model. For example, a detailed explanation for choosing specific methods for semantic prior injection and data augmentation is needed. Providing a comparison with alternative approaches would strengthen the argument.
2. The limitations of using text-to-image models for data augmentation and semantic prior generation could be discussed further. For example, the paper mentions that the Stable Diffusion model doesn't always adhere to the quantifier words in the prompts and that the generated multiple instances often have similar fine-grained attributes. Exploring ways to mitigate these issues or using alternative generative models could be an area for future research.
3. The manuscript could benefit from a more in-depth analysis of the types of queries and images where the proposed method shows significant improvements over the baseline. Understanding the characteristics of the samples that benefit most from the ReSVG approach could provide insights for further enhancements.
4. The computational cost and efficiency of the proposed method compared to the baseline and other state-of-the-art models are not discussed. Providing information on training and inference times, as well as memory requirements, would help readers assess the practicality of the approach for real-world applications.
5. While the proposed method demonstrates strong performance on multiple datasets, it would be interesting to see how well the ReSVG model generalizes to other visual grounding tasks or datasets with different characteristics (e.g., different object categories, more complex spatial relationships, or more diverse query structures). Evaluating the model's robustness and generalization capabilities could strengthen the paper's contributions.
6. The authors could explore the potential of combining the relation-sensitive data augmentation and semantic prior injection techniques with other state-of-the-art visual grounding models beyond TransVG and VLTVG. This would help assess the broader applicability of the proposed methods and their potential to advance the field of visual grounding.

**Suitability:**

2

---

### Official Review · Reviewer_s9cR · 2024-05-23

**Rating:** 4
**Confidence:** 2

**Summary:**

This paper primarily focuses on the challenges related to spatial relationships between objects and fine-grained semantics within the task of visual grounding. To address these challenges, the paper introduces a Relation and Semantic-sensitive Visual Grounding model (ResVG).
Concretely, (1)this work proposes a semantic injection approach with a text2img model to inject text semantics into their model. (2)This work introduces a relation-sensitive data augmentation strategy to synthesize images and corresponding pseudo queries. The experimental results demonstrate the effectiveness of the proposed two components on several datasets.

**Strengths:**

1.The proposed method designs a novel approach to inject text semantic into their model by utilizing a text2img model, which possesses good innovativeness.
2. The manuscript introduces two novel components to improve visual grounding performance and the experimental results indicate the effectiveness of both components.
3. The paper is generally well written and provides useful details for implementation except for the parts I mention below.

**Limitations:**

1. The manuscript lacks an analysis of the model size and efficiency. For instance, there is no discussion on how the introduced text-to-image model affects the inference speed of the overall model.
2. The manuscript lacks some implementation details about the relation-sensitive data augmentation. For instance, (1) if you generate an image with up to 10 objects, how to attach different descriptors (top/bottom/left/right) to each object? (2) which object detector do you choose and if the model chooses a different object detector , will it significantly affect the final performance? (3) some synthesized images might be with low quality, how to identify and deal with such images?

**Suitability:**

2

---

### Official Review · Reviewer_msuz · 2024-05-24

**Rating:** 5
**Confidence:** 3

**Summary:**

This paper focus on a challenging scenario in visual grounding where multiple objects of the same category exist in the image which is always confusing. The paper utilizes recent text-to-image model in two ways to solve the problem: generating reference images as prior information for the queries and generating more relation-sensitive multiple-distraction data to compensate for lacking of data. Both methods are shown to be effective.

**Strengths:**

1. The paper is well-written and easy to follow.
2. The motivation is clear and the proposed method strictly aligns with the motivation.
3. Sec 3.5 provides a detailed discussion of limitations and different possible implementations.
4. The proposed method is validated on various datasets.

**Limitations:**

1. Recent works [1] can generate high-quality detection / grounding dataset using prompts, which can possibly solve the problem that can not control the number of instances and can not generate diverse objects of the same class mentioned in Sec 3.5. I think using data augmentation instead of generating a reference image to enhance the model’s understanding of fine-grained semantics is a more promising solution, as generating a reference image will inevitably increase the inference time.
2. This paper only generates images for predefined classes. Possibly generating more classes beyond predefined classes can further align the text and image space, which may facilitate the understanding of input queries.

[1] Li Y, Liu H, Wu Q, et al. Gligen: Open-set grounded text-to-image generation[C]//Proceedings of the IEEE/CVF Conference on Computer Vision and Pattern Recognition. 2023: 22511-22521.

**Suitability:**

3

---

### Official Review · Reviewer_Xj3d · 2024-05-24

**Rating:** 4
**Confidence:** 3

**Summary:**

The paper proposes to improve the performance of visual grounding when localizing target objects within multiple instance distractions. Through data augmentation and providing visual priors, the problem is mitigated.

**Strengths:**

1. The authors have introduced a novel approach to enhancing visual grounding.
2. Many experiments are performed to demonstrate the effectiveness of the method.
3. The paper is well-structured and easy to read.

**Limitations:**

1. Lack of visualization results.
2. Lack of comparison with the latest works. Visual grounding is also a downstream task for many large vision-language models, such as mPLUG. Maybe more evaluation results should be added.
3. The process of generation is time-consuming.

**Suitability:**

3

---

### Meta-Review · Area_Chair_NFCE · 2024-07-07

**Recommendation:** Accept (Poster)
**Confidence:** 5

**Metareview:**

This paper introduces the Relation and Semantic-sensitive Visual Grounding (ReSVG) model to improve visual grounding in images with multiple-instance distractions. ReSVG enhances fine-grained semantic understanding by using text-to-image generation to inject semantic information from queries. It also introduces a relation-sensitive data augmentation method to generate training data with synthesized images and pseudo queries, addressing the lack of training samples with multiple distractions. These innovations improve the model's comprehension of object semantics and spatial relations, significantly boosting performance in visual grounding tasks, as validated by extensive experiments on five datasets.

Pros
- It is novel to leverage generated images to improve visual grounding.
- Very well-written, organized, and easy to follow.
- Extensive experiments with multiple datasets, ablation studies, and visual examples.

Cons
- The generation process introduces additional complexity, potentially impacting practical application.
- The generative models are trained on abundant data beyond the test task, making it difficult to attribute improvements fairly compared to methods not using generative models.

Overall, this is a decent paper with acceptance scores from all reviewers. The authors are recommended to address the reviewers' comments in the camera-ready version of the paper.